



# Classification of summertime synoptic patterns in Beijing and their association with boundary layer structure affecting aerosol pollution

Yucong Miao[1], Jianping Guo[1], Shuhua Liu[2], Huan Liu[1], Zhanqing Li[3, 4], Wanchun Zhang[1], and Panmao Zhai[1]

[1]State Key Laboratory of Severe Weather & Key Laboratory of Atmospheric Chemistry of CMA, Chinese Academy of Meteorological Sciences, Beijing 100081, China

[2]Department of Atmospheric and Oceanic Sciences, Peking University, Beijing 100871, China

[3]College of Global Change and Earth System Science, Beijing Normal University, Beijing 100875, China

[4]Department of Atmospheric & Oceanic Sciences and ESSIC, University of Maryland, College Park, MD 20740, USA

*Correspondence to*: Jianping Guo, PhD/Prof. (jpguocams@gmail.com)
Zhanqing Li, PhD/Prof. (zli@umd.edu)





**Abstract**

Meteorological conditions within the planetary boundary layer (PBL) are closely governed by large-scale synoptic patterns and play important roles in air quality by directly and indirectly affecting the emission, transport, formation, and deposition of air pollutants. Partly due to the lack of long-term fine-resolution observations of the PBL, the relationships between synoptic patterns, PBL structure, and aerosol pollution in Beijing have not been well understood. This study applied the obliquely rotated principal component analysis in T-mode to classify the summertime synoptic conditions over Beijing using the National Centers for Environmental Prediction reanalysis from 2011 to 2014, and investigated their relationships with PBL structure and aerosol pollution by combining measurements of surface meteorological variables, fine-resolution soundings, the concentration of particles with diameters less than or equal to 2.5 μm, total cloud cover (CLD), and reanalysis data. Among the seven identified synoptic patterns, three types accounted for 67% of the total number of cases studied and were associated with heavy aerosol pollution events. These particular synoptic patterns were characterized by high-pressure systems located to the east or southeast of Beijing at the 925-hPa level, which blocked the air flow seaward, and southerly PBL winds that brought in polluted air from the southern industrial zone. In the vertical dimension, these three synoptic patterns featured a relatively low boundary layer height (BLH) in the afternoon, accompanied by high CLD and southerly cold advection from the seas within the PBL. The cold advection induced by the large-scale synoptic forcing may have cooled the PBL, leading to an increase in near-surface stability and a decrease in the BLH in the afternoon. Moreover, when warm advection appeared simultaneously above the top level of the PBL, the thermal inversion layer capping the PBL may have been strengthened, resulting in further suppression of the PBL and deteriorating aerosol pollution levels. This study has important implications for understanding the crucial roles that meteorological factors (at both synoptic and local scales) play in modulating and forecasting aerosol pollution in Beijing and its surrounding area.




## 1 Introduction

   Beijing, located on the North China Plain (Fig. 1), is the center of politics, culture, and economics in China. With its rapid urbanization, tremendous economic development, and concomitant increase in energy usage, heavy air pollution episodes largely caused by high aerosol loading have been frequently
reported in Beijing (Chan and Yao, 2008; Guo et al., 2011; San Martini et al., 2015; H. Zhang et al., 2015; Zhang and Cao, 2015). The frequent occurrence of pollution events not only reduces visibility and affects transportation, but also has adverse effects on human health (Pope and Dockery, 2006; Chen et al., 2012).

   Great efforts, therefore, have been devoted to investigating air quality issues in Beijing through
comprehensive observational and modelling studies (He et al., 2001; Zhu et al., 2011; Liu et al., 2013; Quan et al., 2013; Zhang et al., 2013; Guo et al., 2014; Wang et al., 2015a; Z. Zhang et al., 2015; Ye et al., 2016). Aerosol pollution has different characteristics in different seasons. In spring, the majority of heavy aerosol pollution is associated with dust storms (He et al., 2001; Zhao et al., 2007; Guo et al., 2013). During summer and fall, photochemical production and agricultural burning may play a role in
exacerbating the air quality (Zhang et al., 2013; Wang et al., 2015b). In winter, frequent severe haze events were found to be associated with the substantial increase in coal combustion for heating (Zhang et al., 2013; Tang et al., 2015). Air quality can be further deteriorated through secondary aerosol formation (Huang et al., 2014; Sun et al., 2014; Han et al., 2015).

   In addition to high emissions and aerosol chemistry processes, meteorological conditions also play
important roles in the formation and evolution of aerosol pollution in Beijing (Zhang et al., 2012; Quan et al., 2013; Hu et al., 2014; Miao et al., 2015b; Ye et al., 2016; Z. Zhang et al., 2015). In the heavily populated monsoon region (e.g., Beijing), the large-scale distributions and variations of aerosol loading are strongly influenced by the monsoon circulations (Zhang et al., 2010; Li et al, 2016; Wu et al., 2016), since the monsoon circulations directly determine the large-scale transport and lifetime of aerosols. In
addition to the increase of anthropogenic emissions during past decades in China, the recently frequent occurrence of aerosol pollution in China has been found to be associated with the variability in Monsoon (Niu et al., 2010; Zhang et al., 2010; Liu et al., 2011). Based on meteorological visibility data, a significantly negative correlation between aerosol pollution in China and winter monsoon were





revealed on inter-annual timescale (Chen and Wang, 2015; Qu et al., 2015; Li et al., 2015). Besides, a weak eastern Asian summer monsoon is also found to be accompanied with high summer surface layer aerosol concentrations in northern China (Zhang et al., 2010; Liu et al., 2011; Yan et al., 2011; Zhu et al., 2012).

In the local scale, detailed statistical analyses have also shown that heavy aerosol pollution in Beijing is associated with southerly winds, high relative humidity (RH), stable atmospheric stratification, and a low boundary layer height (BLH) (Quan et al., 2013; H. Zhang et al., 2015; Tang et al., 2016; Ye et al., 2016). Among these local meteorological factors, the BLH is one of the most crucial factors because it directly determines the total dispersion volume (Stull, 1988; Quan et al., 2013; Hu et

al., 2014; Tang et al., 2016) and the structure of the planetary boundary layer (PBL). The latter has a strong influence on the occurrence, maintenance, and dissipation of aerosol pollution in Beijing (Miao et al., 2015b; Ding et al., 2016; Petäjä et al., 2016; Tang et al., 2016).

With the Yan and Taihang Mountains to the north and west of Beijing (Fig. 1), the thermally-induced mountain-plain breeze circulation can be well developed under favorable synoptic conditions,

which impacts the PBL structure and modulates aerosol pollution (De Wekker, 2008; Chen et al., 2009; Liu et al., 2009; Hu et al., 2014; Miao et al., 2015b, 2016). Since the Bohai Sea is located ~150 km to the southeast of China (Fig. 1), a sea-breeze (Miller, 2003) can be established and penetrate inland to Beijing. This can also play a role in affecting the PBL structure and air quality (Liu et al., 2009; Sun et al., 2013; Miao et al., 2015a, 2015b).

Although the importance of the PBL structure on air pollution has been widely recognized (Wang et al., 2014; Miao et al., 2015b; Tang et al., 2016), more investigations are warranted concerning (1) the crucial factors affecting the development of the PBL and (2) the relationships between large-scale synoptic forcings and the structure of the PBL. They are yet to be fully understood, partly due to the lack of long-term fine-resolution PBL observations (Liu and Liang, 2010). Radiosondes are

conventionally launched twice a day at 0000 (0800) and 1200 (2000) Coordinated Universal Time (UTC) (Beijing Local Time, BJT). Most data are only reported at significant pressure levels with at





most six records below 500 hPa (Liu and Liang, 2010), which cannot capture the fine structure of the PBL.

In 2011, an L-band radiosonde network across China (Guo et al., 2016; Zhang et al., 2016) was established by the China Meteorological Administration. This network of measurements provides fine-

resolution profiles of temperature (Fig. 2), pressure, RH, and wind speed and direction twice a day (0800 and 2000 BJT). Additional soundings are made at 1400 BJT in the summertime (June-July-August, the wet season) at the Beijing site (39.80°N, 116.47°E) (the blue cross in Fig. 1). These fine-resolution sounding observations allow for the investigation of the PBL structure over Beijing.

The development of the PBL is mainly controlled by large-scale external synoptic conditions and by

local surface sensible heat fluxes (Garratt, 1994). So in this study, not only are local factors that affect the surface heat budget analyzed (e.g., cloud cover), but also the large-scale synoptic forcing. A synoptic regime defined by large-scale warm/cold advection and transport pathways of water vapor and pollutants (Zhang et al., 2012; Zhao et al., 2013; Ye et al., 2016) can affect the PBL structure and air quality (Miao et al., 2015b; Ye et al., 2016). For example, severe aerosol pollution events in Beijing

have occurred more frequently under stable and weak anticyclone synoptic conditions (Zhang et al., 2012; Liu et al., 2013; Li et al., 2015).

In this study, we employ a climatological approach to classify the summertime synoptic patterns over Beijing from 2011 to 2014, and unravel their relations to the PBL structure and aerosol pollution. Most previous studies evaluated the impacts of synoptic types through case studies for short periods

(Chen et al., 2009; Hu et al., 2014; Quan et al., 2014; Tie et al., 2015; Miao et al., 2016). This approach is not suitable for identifying dominant synoptic patterns. Studying large-scale synoptic patterns allows us to consider the numerous interrelated meteorological variables within an integrated framework (Zhang et al., 2012), thus providing an insight into the physical mechanisms underlying aerosol pollution in Beijing.

Approaches used to classify synoptic patterns can be roughly split into two groups: subjective and objective. The subjective approach is usually referred to as manual classification, which is subjectively defined a priori and where the case assignment is also subjective (Huth et al., 2008). A subjective





classification is arbitrary to a large extent. By contrast, an objective approach defines types and assigns cases using numerical procedures based on the measures of (dis)similarity and variance maximization. Because an objective classification is capable of processing large amounts of data and depends less on one's experience, we choose this objective approach (Huth et al., 2008) to identify synoptic types in

Beijing.

The same classification approach was applied by Zhang et al. (2012) and Ye et al. (2016) to classify synoptic types in the North China Plain. Zhang et al. (2012) identified nine synoptic patterns using the daily surface-level pressure fields from 2000 to 2009, and Ye et al. (2016) presented a more detailed classification of synoptic patterns for fall and winter. They found that the poor air quality in Beijing was

associated with high pressure to the east and a relatively low BLH. However, previous studies did not unravel the causes of the low BLH and the physical mechanisms underlying it, partly due to the lack of appropriate observational data (e.g., fine-resolution soundings in the afternoon, cloud cover). In addition, the north-south movement of the subtropical anticyclone plays an important role in modulating the seasonal variation in prevailing synoptic patterns in Beijing (Miao et al., 2015b). It is thus better to

classify synoptic patterns for each season.

In summer, although the aerosol pollution level in Beijing is lower than that in fall and winter (Fig. 3a), the seasonally average concentration of particles with diameters less than or equal to 2.5 μm ($PM_{2.5}$) is still as high as 85.7 μg m$^{-3}$, based on data from 2011 to 2014. This is 2.4 times higher than the national standard level (35 μg m$^{-3}$). Also, the relationships between aerosol pollution, the PBL structure,

and synoptic patterns in summer are rarely studied. Here, we used an objective approach to classify summertime synoptic patterns in Beijing, and evaluated the relationships of these synoptic patterns with aerosol pollution and PBL structures using long-term observations. This will extend previous studies as it is an attempt to understand the impacts of large-scale synoptic forcings on the PBL structure and air quality.

The remainder of this paper is organized as follows. In section 2, the methodology and data are described. In section 3, the summertime synoptic types are classified, and their relationships to aerosol pollution and the PBL structure are investigated. The main findings are summarized in section 4.





## 2 Data and methods

### 2.1 Data

To classify the summertime synoptic types in Beijing, geopotential height (GH) fields derived from

the National Centre for Environmental Prediction (NCEP) global Final (FNL) reanalysis
(http://rda.ucar.edu/datasets/ds083.2/) from 2011 to 2014 are used. The NCEP-FNL reanalysis is
produced by the Global Data Assimilation System, which continuously assimilates observations from
the Global Telecommunication System and other sources. The NCEP-FNL reanalysis fields are on 1° ×
1° grids with a 6-hour temporal resolution, i.e., 0000 (0800), 0600 (1400), 1200 (2000), and 2000

(0200) UTC (BJT).

In this study, daily GH fields at the 925-hPa level from the NCEP-FNL reanalysis covering the
Beijing-Tianjin-Hebei (BTH) region (43°N-47°N, 112°E-125°E) (Fig. 1) were classified to identify the
prevailing synoptic types in summer. Results from the classification of the 925-hPa GH field are similar
to those using the GH fields at other tropospheric levels because there is a high degree of dependence

among individual levels (Huth et al., 2008).

To investigate the PBL structures associated with different synoptic types, summertime soundings
collected at the Beijing site (39.80°N, 116.47°E) for the period 2011-2014 were analyzed. This
sounding station, equipped with an L-band radiosonde system (Guo et al., 2016), provides atmospheric
sounding data (profiles of temperature, RH, wind speed and direction) up to three times a day (0800,

1400, and 2000 BJT) at a high vertical resolution. In total, 1055 effective soundings were obtained for
this study:357 soundings at 0800 BJT, 332 soundings at 1400 BJT, and 366 soundings at 2000 BJT. In
addition, the total cloud cover (CLD) was observed four times a day (0200, 0800, 1400, and 2000 BJT)
and hourly near-surface observations (temperature, RH, wind speed and direction, and precipitation
amount) were made.

The aerosol pollution level in Beijing is denoted by the near-surface $PM_{2.5}$ concentration. Since
2008, U.S. diplomatic missions in China have monitored $PM_{2.5}$ concentrations and have made both real-
time and historic data available to the public (www.stateair.net). Hourly $PM_{2.5}$ concentrations in Beijing
(the red dot in Fig. 1) are measured using a beta attenuation monitor (BAM) (Chung et al., 2001)





installed on the roof of the U.S. Embassy (39.95°N, 116.47°E). The BAM technique is a reference method for measuring $PM_{2.5}$ concentrations that is used by the U.S. Environmental Protection Agency. In this study, summertime $PM_{2.5}$ concentration measurements from 2011 to 2014 were used to investigate the relationship between aerosol pollution and synoptic patterns in Beijing.

## 2.2 BLH derived from soundings

The bulk Richardson number (Ri) method (Vogelezang and Holtslag, 1996) was applied to estimate the BLH in Beijing from sounding data because it is suitable for both stable and convective PBLs (Seidel et al., 2012). The Ri is defined as the ratio of turbulence associated with buoyancy to the

turbulence caused by mechanical shear:

$$\mathrm{Ri}(z) = \frac{(g/\theta_{vs})(\theta_{vz}-\theta_{vs})(z-z_s)}{(u_z-u_s)^2+(v_z-v_s)^2+bu_*^2} u_* \quad , \quad (1)$$

where $z$ is the height (above ground level, AGL), $g$ is the acceleration caused by gravity, $\theta_v$ is the virtual potential temperature, $u$ and $v$ are the components of the observed wind speed, $b$ is a constant, and $u_*$ is the surface friction velocity. The subscript $s$ denotes the surface level. Since $u_*$ is not known from the

sounding observations, and its magnitude is much smaller than that of the bulk wind shear term in the denominator (Vogelezang and Holtslag, 1996), we set b = 0 and ignore the surface frictional effect. The BLH is referred to as the lowest level $z$ at which the interpolated Ri crosses the critical value of 0.25. A similar criterion was applied to investigate PBL climatologies by Seidel et al. (2012) for the U.S. and by Guo et al. (2016) for China. A case in point for the BLH derived from the sounding profiles of $\theta_v$ and Ri

at the Beijing site is shown in Fig. 2.

## 2.3 Classification of synoptic types

The obliquely rotated principal component analysis in T-mode (T-PCA) approach (Richman, 1981; Huth et al., 2008) was first used to analyze large-scale synoptic conditions through classification of the

predominant synoptic types, which was adopted for air quality studies (e.g., Stefan et al., 2010; Zhang et al., 2012). The T-PCA calculates the eigenvectors of the input data set by singular value





decomposition and finds typical patterns by loadings that can be divided into classes. The application of
the PCA in T-mode means that daily patterns form the columns in the input data matrix and grid-point
values form its rows (Huth, 2000). Unlike the common application of PCA is in the S-mode, which is
used to isolate subgroups of grid points that co-vary similarly, the T-PCA is used to isolate subgroups of

similar spatial patterns. This approach has proven to be a reliable classification method, largely due to
its temporal and spatial stability, in addition to its ability to reproduce predefined dominant patterns
with little dependence on pre-set parameters (Huth, 1996; Philipp et al., 2010).

In this study, the T-PCA classification based on Huth (2000) was done using the cost733class
software package (http://cost733.met.no), which was developed for creating, comparing, and evaluating

classifications in several variants. Using the T-PCA module of the cost733class software package to
classify synoptic types, the number of principal components needs to be explicitly defined. To
understand the relationships between synoptic types, PBL structures, and aerosol pollution, we tested
the classification using different numbers of principal components (e.g., 4, 5, 6, 7, 8, 9), and compared
the results with the observed BLH and $PM_{2.5}$ concentration (Fig. S1). The classification using seven

principal components shows the most significant anti-correlation between BLH and $PM_{2.5}$ concentration
($R = -0.97$, $p < 0.01$, Fig. S1). This will be discussed further in section 3.

## 3 Results and discussion

3.1 Relationships between the BLH and aerosol pollution in summer

During a diurnal cycle, the development of the PBL is closely tied to solar heating of the ground
(Stull, 1988). After sunrise, the PBL in Beijing during summer undergoes a transition from a nocturnal
stable PBL to a convective PBL, and reaches its maximum depth in the afternoon (c.f., Fig. 3b),
remaining there until sunset (Stull, 1988). After sunset, without sufficient heat fluxes to maintain the

convective PBL, the BLH drops quickly (Fig. 3b). Along with the diurnal evolution of the BLH, the
near-surface $PM_{2.5}$ concentration exhibits a nearly reversed diurnal variation. The $PM_{2.5}$ concentration is
generally low in the afternoon and reaches its peak values in the evening and early morning.



As shown in Fig. 4, the BLHs derived from soundings were compared with their corresponding daily average $PM_{2.5}$ concentrations. At 0800 BJT, the PBL is typically shallow and its depth is statistically uncorrelated with the daily $PM_{2.5}$ concentration (R = -0.03, p = 0.62, Fig. 4a). At 1400 BJT, the PBL is fully developed, with an average BLH of ~1.3 km, which is clearly anti-correlated with the

daily $PM_{2.5}$ concentration (R = -0.36, p < 0.01, Fig. 4b). This implies that the variation in afternoon BLH plays an important role in modulating the variation in aerosol pollution level in Beijing. At 2000 BJT, the PBL is in transition from a convective state to a nocturnal stable state, and the BLH becomes uncorrelated with the daily $PM_{2.5}$ concentration (R = 0.10, p = 0.09, Fig. 4c). Averaged over the whole day, a significantly negative correlation (R = -0.32, p < 0.01) is still found between the daily BLH and

$PM_{2.5}$ concentration (Fig. 4d). When excluding observations made on rainy days, a stronger correlation (R = -0.37, p < 0.01) is obtained. So in this study, we mainly investigate the factors that determine the BLH at 1400 BJT and their relationships with large-scale synoptic patterns.

3.2 Synoptic patterns and aerosol pollution

Using the T-PCA classification based on the summertime 925-hPa GH fields, seven dominant types of synoptic patterns (Fig. 5 and Table S1) were identified. According to the locations of high and low pressure systems with respect to Beijing (Fig. 5), these seven synoptic types can be briefly described as (1) high pressure to the east, (2) high pressure to the north, (3) low pressure to the northeast, (4) low pressure to the north, (5) low pressure to the northwest, (6) weak high pressure over Beijing, and (7)

low pressure to the east.

Among these seven identified synoptic patterns, Types 1, 4, and 5 are associated with heavier aerosol pollution (Fig. 5), all with an average $PM_{2.5}$ concentration greater than 90 µg m$^{-3}$. These three types are also the most frequent synoptic patterns in summer, accounting for 67% of the total. At the 925-hPa level, the three synoptic patterns are characterized by a high pressure system located to the east

or southeast of Beijing, which brings southerly winds to Beijing, but blocks the polluted air from moving away eastwards towards the Yellow Sea (Figs. 5 and S2). Southerly winds are not only observed at the 925-hPa level, but also throughout almost all of the PBL (Fig. 6a). With the southerly PBL winds blowing over the plains of the BTH region, pollutants emitted from the surrounding





southern cities (e.g., Baoding, Shijiazhuang, and Cangzhou) can be transported to Beijing (Wang et al., 2010; Zhang et al., 2012; Miao et al., 2016), leading to a worsening of the air quality in Beijing.

In addition to the southerly PBL winds, high 2-m RH (RH2), high CLD, and low BLH are other crucial parameters associated with severe aerosol pollution (Figs. 6 and 7). At 1400 BJT, all three
synoptic patterns (Types 1, 4, and 5) have a relatively low BLH (Fig. 6a), with an average value less than 1.4 km. This would limit the vertical dispersion of pollutants, leading to high $PM_{2.5}$ concentrations. Meanwhile, Types 1 and 5 are characterized by relatively high CLD and RH2 (Fig. 6b), which could facilitate the hygroscopic growth of aerosols (Kim et al., 2014; Ye et al., 2016).

As illustrated in Fig. 7, RH2, CLD, and BLH are highly related. The increase in CLD reduces the
amount of solar radiation reaching the surface, and suppresses the development of the PBL and the vertical mixing of water vapor, leading to a decrease in the BLH and an increase in RH2. On the other hand, when RH2 increases, the lifting condensational level (LCL) can lower, favoring the formation of cumulus clouds, which may subsequently suppress the development of the PBL (Wilde et al., 1985; Craven et al., 2002; Zhu and Albrecht, 2002). As a result, under clear conditions (CLD < 20%), the
average RH2 is less than 40% while the average BLH at 1400 BJT can reach ~2.2 km (Fig. 7), favoring the vertical dispersion of aerosols. By contrast, under cloudy conditions (CLD > 80%), the average RH2 and BLH increases to ~70% and decreases to ~1.2 km, respectively.

Figure 8 shows the correlations between average $PM_{2.5}$ concentrations and meteorological variables for the different synoptic patterns. The $PM_{2.5}$ concentration in Beijing is significantly correlated with the
southerly wind at the 925-hPa level, RH2, and CLD at 1400 BJT, and anti-correlated with the BLH at 1400 BJT. By contrast, the $PM_{2.5}$ concentration is uncorrelated with the 2-m temperature and wind speed. Types 1, 4, and 5 synoptic patterns associated with southerly PBL winds, high CLD, low BLH, and high RH2 in Beijing favor the occurrence of heavy aerosol pollution in summer. By contrast, synoptic patterns with strong northerly PBL winds (Types 3 and 7) or relatively high BLH (Types 2 and
6) are associated with good air quality conditions in Beijing.

In addition to the aforementioned meteorological factors, precipitation may also affect aerosol pollution levels through wet scavenging (Yoo et al., 2014). Therefore, for each identified synoptic





pattern, we examined the aerosol pollution after removing observations made on rainy days (Figs. S3 and S4). Similar relationships between aerosol pollution, RH2, CLD, BLH, and synoptic patterns were found, suggesting that whether it rains or not is not important when it comes to understanding effects of synoptic patterns on PBL structure and aerosol pollution in Beijing.

### 3.3 Large-scale synoptic warm/cold advection and PBL structures

As illustrated Fig. 8, from the perspective of synoptic patterns, the BLH is the most crucial factor related to the aerosol pollution level under different synoptic conditions ($R = -0.97$, $p < 0.01$). In addition to CLD and RH2, the BLH is also highly related to the southerly wind at the 925-hPa level,

implying that the southerly synoptic advection may also play a role in modulating the development of the PBL.

To understand how large-scale synoptic advection affects the development of the PBL in Beijing in the presence of different synoptic patterns, the three-dimensional temperature and wind vector fields derived from the NCEP-FNL reanalysis were analyzed and compared with sounding data. As the

potential temperature (PT) and wind vector profiles illustrate in Fig. 6a and Fig. 9, although the NCEP-FNL reanalysis tends to overestimate PT within the PBL and some discrepancies in the wind vector profiles are seen, the general characteristics of the PT and wind vector profiles of each synoptic pattern are captured well by the reanalysis, such as the sharp changes in PT and the wind vector across the top level of the PBL. The good agreement between the soundings and the NCEP-FNL reanalysis provides a

basis for using the reanalysis to understand the relationships between large-scale synoptic forcings and PBL structures.

The spatial distributions of the PT anomaly (subtracted from the average PT over the whole study region) at the 925-hPa level at 1400 BJT are shown in Fig. 10. In the afternoon, the near-surface air over the sea (located to the southeast of the study region) is cooler and wetter than that over the plains.

So when southerly and easterly PBL winds appear over the southeast coastal regions, the cool marine air can be advected to Beijing (Fig. 10) and impose a negative thermal anomaly to the PBL there (Fig. 11).



Among the seven identified synoptic patterns, the strongest near-surface cold advection is associated with Type 1 (Fig. 11a), leading to the coldest PBL at 1400 BJT (Fig. 9a). In the vertical dimension, the cold anomaly is stronger within the PBL than at the upper level (Fig. 11a). Such a differential cooling effect increases the thermal contrast between the PBL and its upper level, and

strengthens the thermal inversion there, leading to suppression of the development of the PBL. Types 2, 4, 5, and 6 also show cold advection toward Beijing but it is less prominent (Figs. 11b and 11d-f). For example, at 1400 BJT, the cold advection seen in Types 4 and 6 hardly reaches Beijing (Figs. 11d and 11g).

On the western side of the BTH region, mountains act as elevated heat sources in the afternoon and

warm the air adjacent to the mountains (Figs. 11 and 12). In the presence of northerly or westerly synoptic winds above the mountains, the warmer air above the mountains may be transported to the plain regions of Beijing (Figs. 11 and 12). This prominent warm advection above the PBL is seen in Types 3, 4, 5, 6, and 7 (Figs. 11c-g and 12c-g). Among these five synoptic types, except for Type 6 where the northwesterly synoptic wind appears at a relatively high level (~750 hPa, Fig. 6a), the

northerly or westerly synoptic winds associated with the other four types are observed around the top level of the PBL.

For Types 4 and 5, the simultaneous presence of warm advection at the PBL top and cold advection within the PBL could increase the thermal stability of the PBL, strengthening the thermal inversion capping the PBL top and suppressing the growth of the BLH (Figs. 11d-f). With a relatively strong cold

advection in Type 5, the BLH over Beijing is dramatically lowered to ~1 km AGL at 1400 BJT (Figs. 9e and 11e), which is the lowest BLH among all seven identified synoptic patterns. By contrast, for Types 3 and 7, the PBL over Beijing is only affected by the warm advection aloft in the afternoon (Figs. 11c and 11g), which could reach ~1.7 km AGL, favoring the vertical dispersion of PBL pollutants.

In short, in the afternoon, the cold advection within the PBL, induced by large-scale synoptic winds,

could decrease the PBL temperature and strengthen the thermal inversion capping the PBL, thus suppressing its development. In addition, the simultaneous presence of warm advection at the PBL top could further strengthen the thermal inversion and intensify the suppression of the PBL. Such a


mechanism could be partially responsible for the relatively low BLH of Types 1, 4, and 5 at 1400 BJT, leading to the high aerosol concentrations at the surface level then. With favorable thermal conditions in the summertime, the local atmospheric circulations (e.g., mountain-plain breeze, sea-land breeze) may be in place and superimposed on the large-scale synoptic advection, influencing the PBL structure in

Beijing (Hu et al., 2014; Miao et al., 2015b).

### 4 Conclusions

In this study, seven different synoptic patterns during the summer in Beijing were identified using the T-PCA method and NCEP-FNL reanalysis data from 2011 to 2014. Their relationships with aerosol

pollution and the PBL structure were comprehensively investigated using collocated long-term $PM_{2.5}$ measurements and fine-resolution soundings in Beijing.

The climatological diurnal cycle of the BLH, revealed by thrice-daily soundings (at 0800, 1400, and 2000 BJT) shows that the BLH reaches a maximum in the afternoon and is anti-correlated with the near-surface $PM_{2.5}$ concentration. In addition, daily aerosol pollution is significantly anti-correlated with

the BLH at 1400 BJT. The correlation analysis between the BLH and local meteorological parameters shows that in the afternoon, the BLH is negatively correlated with CLD and RH2. Thus, heavy aerosol pollution events frequently occur under cloudy conditions with high RH2 and low BLH.

By classifying the summertime 925-hPa GH fields over the Beijing-Tianjin-Hebei region (43$^o$N-47°N, 112$^o$E-125°E), three types of synoptic patterns favoring the occurrence of heavy aerosol pollution

were identified, accounting for 67% of the total. At the 925-hPa level, these three synoptic types were characterized by high pressure systems located to the east or southeast of Beijing, leading to southerly PBL winds in Beijing. These southerly PBL winds favor the transport of pollutants from the surrounding southern industrial cities (e.g., Baoding, Shijiazhuang, and Cangzhou) to Beijing. In the vertical dimension, these three synoptic types were characterized by a relatively low BLH at 1400 BJT,

which may be caused by the relatively high CLD.

In addition, the large-scale atmospheric advection within the PBL may play an important role in modulating the development of the PBL. In the afternoon, cold advection from the south within the PBL,





induced by large-scale synoptic forcings, could decrease the PBL temperature and strengthen the thermal inversion above the PBL, thus suppressing the development of the PBL. Moreover, when warm advection appears simultaneously at the top level of the PBL, the suppression of the PBL is intensified. Such a mechanism is summarized in the schematic diagram shown in Fig. 13.

Although the important impacts of large-scale synoptic forcing on PBL structures and air quality in Beijing during the summer have been emphasized in this study, the important roles of local atmospheric circulations on the modulation of the PBL structure and air pollution cannot be overlooked. The presence of aerosols may also play a role in modifying PBL structures and processes, which warrants further study and will form the basis of our future work.

## Acknowledgements

This study is supported by the National Natural Science Foundation of China (91544217 and 41471301), the Ministry of Science and Technology of China (2014BAC16B01), and the Chinese Academy of Meteorological Sciences (2014R18). The authors would like to acknowledge the China

Meteorological Administration for providing the long-term sounding and total cloud cover data, and the U.S. diplomatic missions for providing the $PM_{2.5}$ concentration measurements.

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



**Figures**

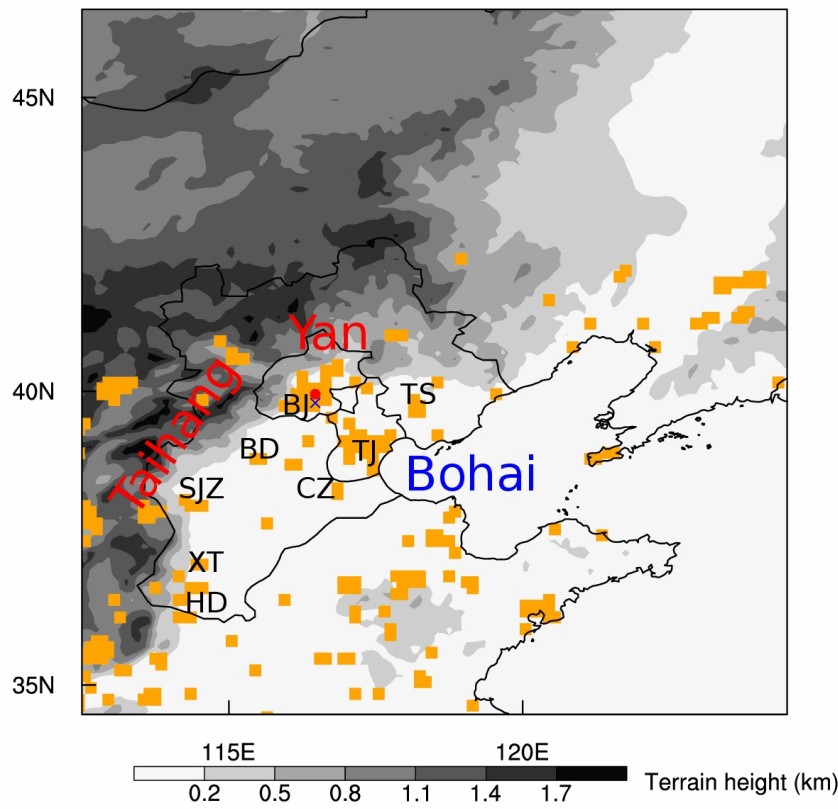

**Fig. 1.** Spatial distribution of terrain height of the North China Plain. Superimposed are the locations of
5   the U.S. Embassy air quality station (39.95°N, 116.47°E, the red dot) and the Beijing meteorological
station (39.80°N, 116.47°E, the blue cross). Urbanized areas based on MODIS 2012 data are shown in
orange. The locations of Beijing (BJ) and adjacent industrial cities are written in black text and include
Tianjin (TJ), Tangshan (TS), Baoding (BD), Cangzhou (CZ), Shijiazhuang (SJZ), Xingtai (XT), and
Handan (HD).




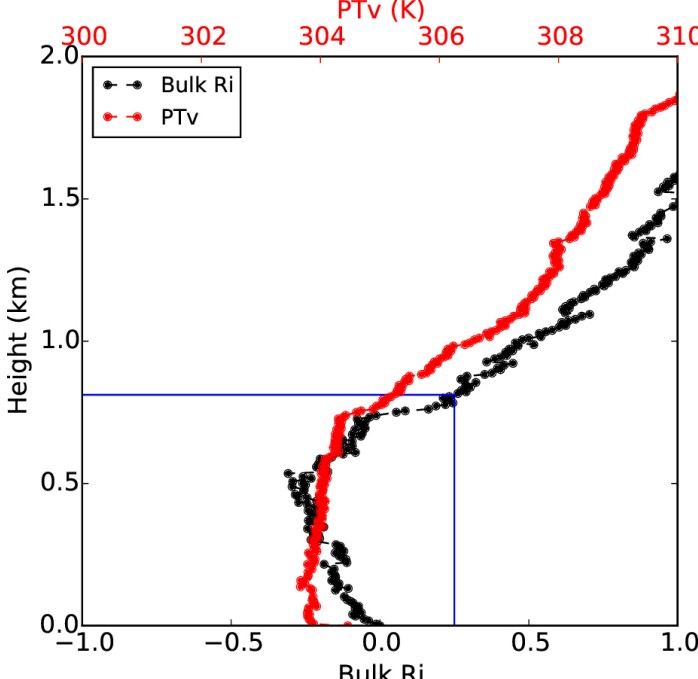

**Fig. 2.** Vertical profiles of the bulk Richardson number (Ri, in black) and virtual potential temperature (PTv, in red) based on L-band sounding observations made in Beijing on 30 June 2013 at 1400 BJT. The boundary layer height is the height where Ri first reaches the value of 0.25 (the blue lines).





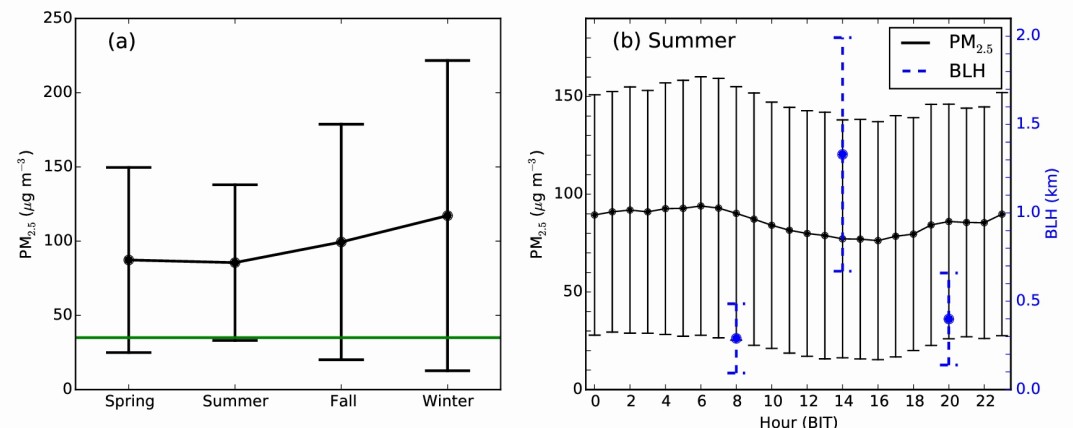

**Fig. 3.** (a) Seasonal variation of the daily averaged $PM_{2.5}$ concentration (the mean ± one standard deviation), and (b) diurnal cycle of the $PM_{2.5}$ concentration (the mean ± one standard deviation) in summer derived from hourly measurements made at the U.S. Embassy air quality station from 2011 to 2014. The green line in (a) shows the national standard level (35 μg m$^{-3}$). The diurnal cycle of the BLH (in blue, the mean ± one standard deviation) derived from summertime soundings in Beijing is shown in (b) on the right-hand ordinate.



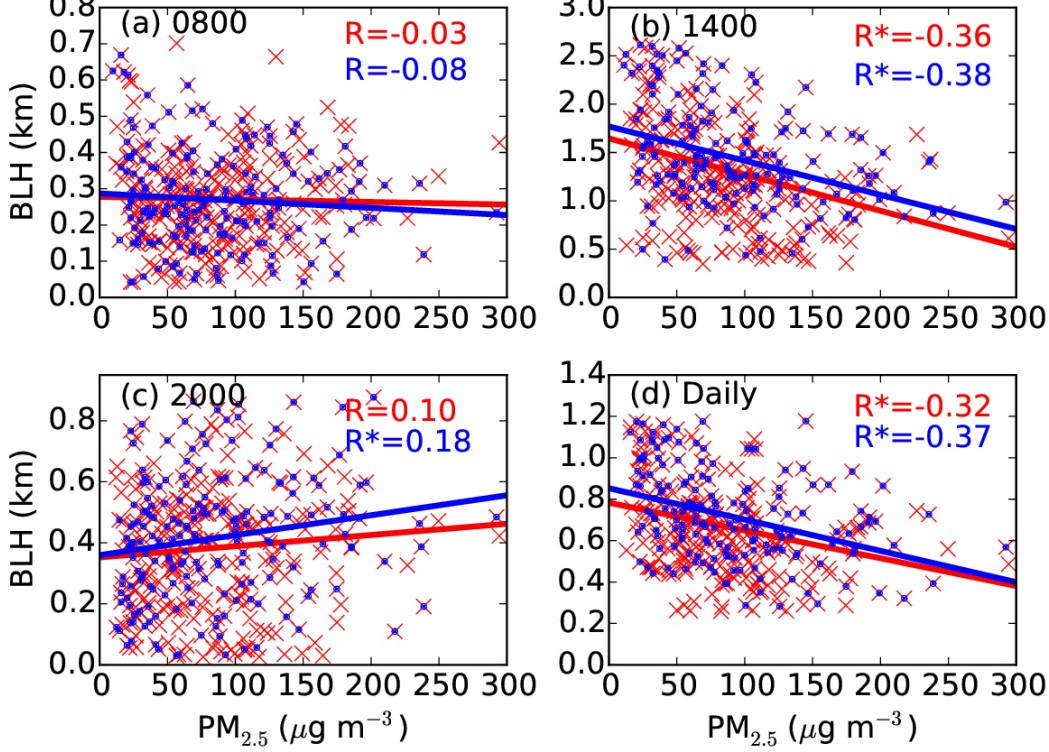

Fig. 4. The BLH as a function of daily $PM_{2.5}$ concentration at (a) 0800 BJT, (b) 1400 BJT, (c) 2000 BJT, and (d) averaged over the whole day, in red. BLHs greater than the 95[th] percentile value and less than the 5[th] percentile value are not included in the plots. The correlations between the $PM_{2.5}$ concentration and BLH without considering measurements made on rainy days are shown in blue. The correlation coefficients (R) are given in each panel, and the asterisks indicate values that are statistically significant ($p < 0.05$).





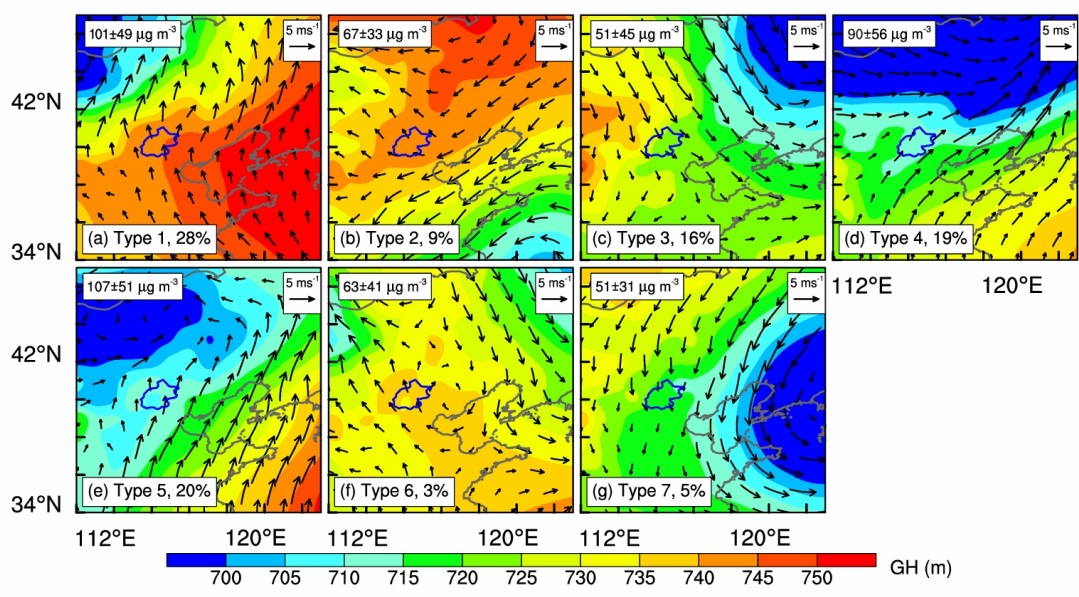

**Fig. 5.** 925-hPa geopotential height (GH) fields (colored areas) and wind vector fields (arrows) in summer from 2011 to 2014 for the seven synoptic patterns: (a) Type 1, (b) Type 2, (c) Type 3, (d) Type 4, (e) Type 5, (f) Type 6, and (g) Type 7. The occurrence frequency of each synoptic pattern is given in the bottom left of each panel and the PM$_{2.5}$ concentration (the mean ± one standard deviation) is shown in the top left of each panel. The location of the Beijing metropolitan area is outlined in blue near the center of each panel.





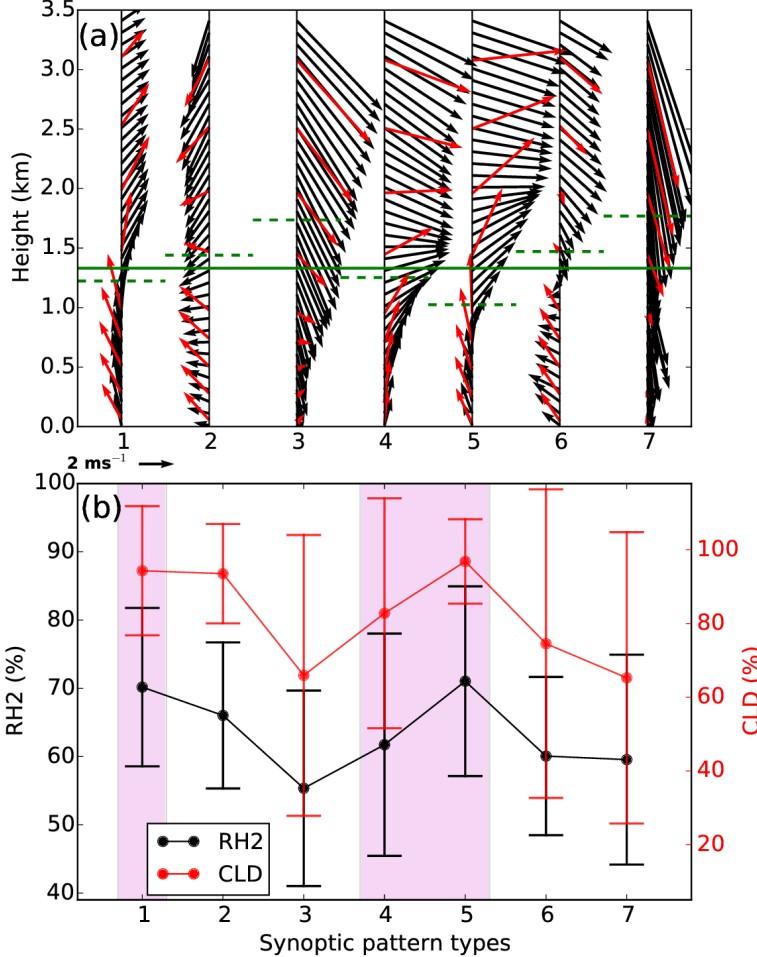

**Fig. 6.** (a) Average wind vector profiles at 1400 BJT and (b) observed daily RH2 and CLD at 1400 BJT (mean ± one standard deviation) associated with the seven types of synoptic pattern. In (a), the wind vectors derived from soundings are denoted by black vectors and those from the NCEP-FNL reanalysis are in red. The green solid line shows the seasonal mean BLH and the dashed green lines represent the mean BLH for each synoptic pattern type. Note that the wind vector profiles from the NCEP-FNL reanalysis were derived by interpolating the values from the four nearest grids.





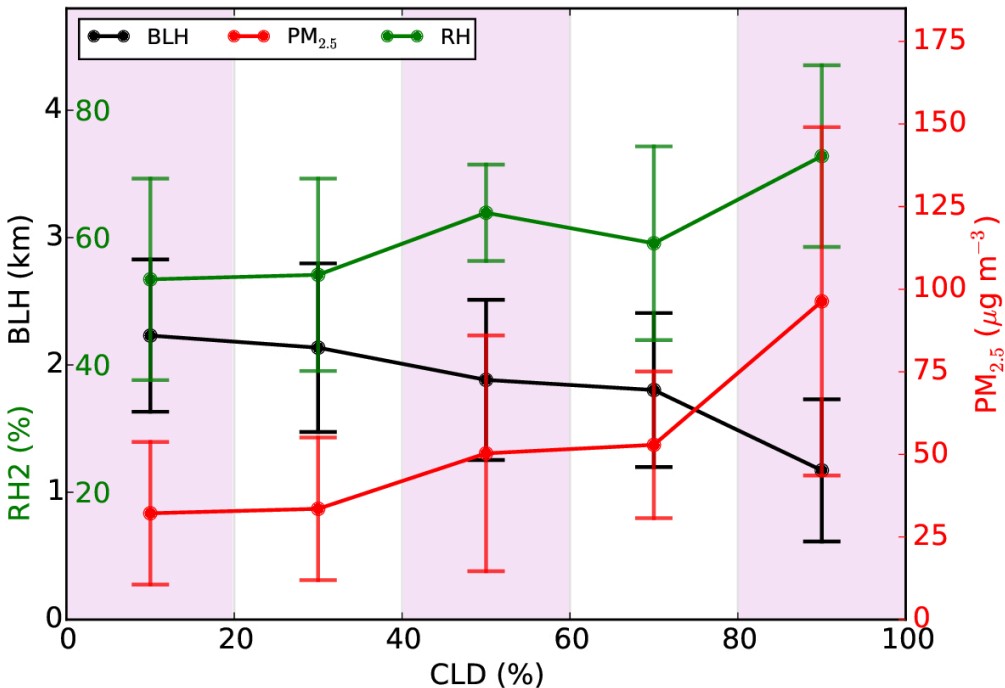

**Fig. 7.** The BLH at 1400 BJT (in black), daily RH2 (in green), and daily PM$_{2.5}$ concentration (in red) as a function of CLD at 1400 BJT. Mean values ± one standard deviation are shown. The bin size is 20%.





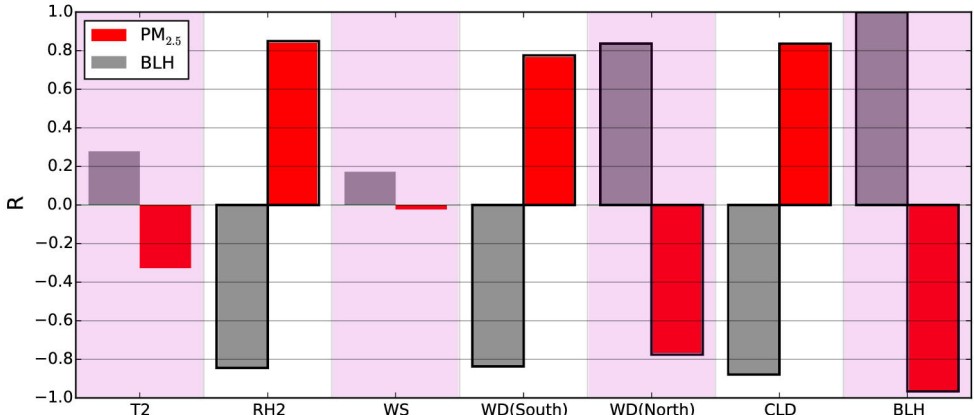

**Fig. 8.** Correlations (R) between the mean values of $PM_{2.5}$ concentration and meteorological parameters for the different synoptic patterns, including (from left to right) 2-m temperature (T2), 2-m relative humidity (RH2), wind speed at the 925-hPa level (WS), south- and north- wind frequencies at the 925h-hPa level (WD), total cloud cover at 1400 BJT (CLD), and the BLH at 1400 BJT. The grey bars represent the correlations between BLH and these meteorological parameters. Bars outlined in thick black lines indicate correlation coefficients (R) that are statistically significant ($p < 0.05$).





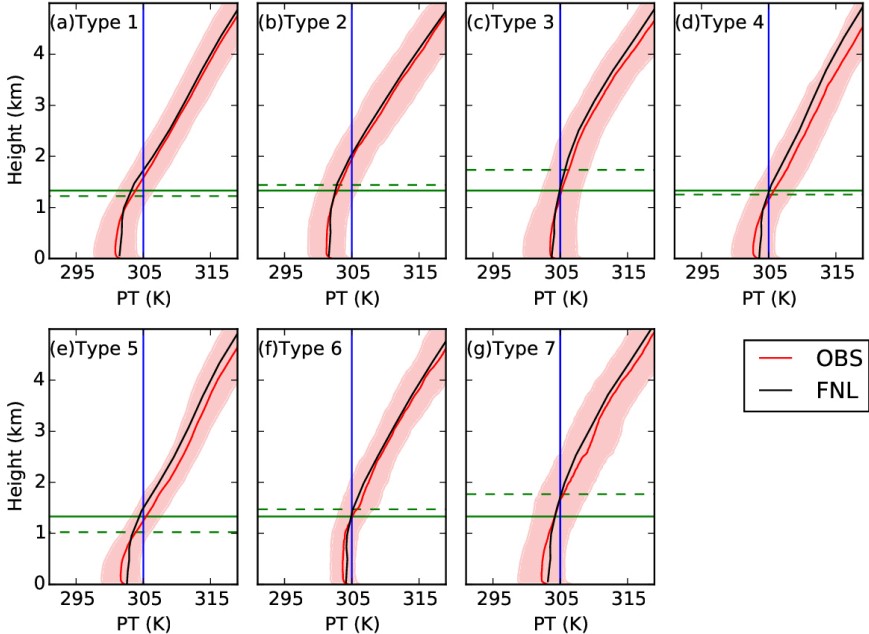

Fig. 9. Vertical profiles of potential temperature (PT) at 1400 BJT associated with the seven types of synoptic pattern derived from soundings (red lines) and the NCEP-FNL reanalysis (black lines). Solid lines indicate average values and shaded areas show the uncertainty range (the mean ± one standard
5    deviation). Green solid lines represent the seasonally averaged BLH and green dashed lines represent the average BLH for each synoptic type. The PT profiles from the NCEP-FNL reanalysis were derived by interpolating the values from the four nearest grids.





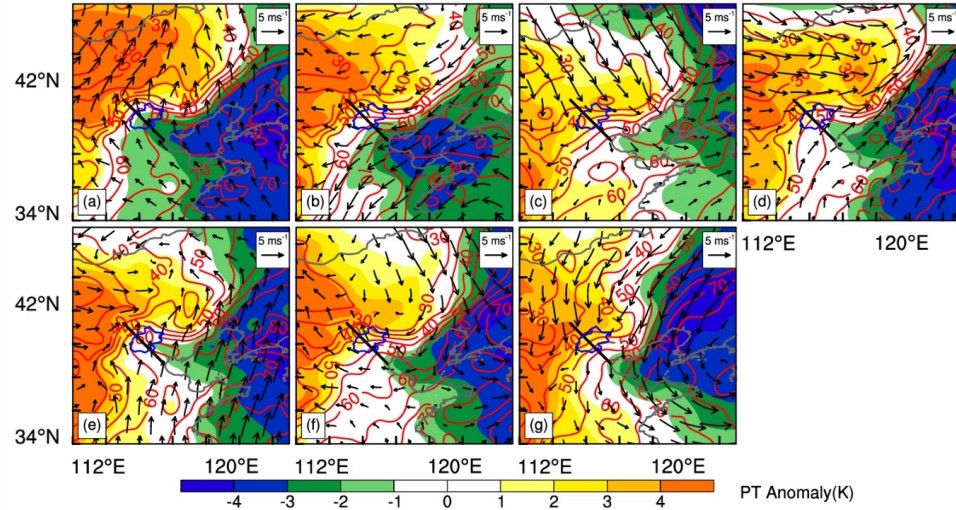

**Fig. 10.** Spatial distributions of the 925-hPa PT anomaly (subtracted from the average PT over the whole study region, colored areas), wind vectors (arrows), and RH (red contour lines) for the seven synoptic pattern types: (a) Type 1, (b) Type 2, (c) Type 3, (d) Type 4, (e) Type 5, (f) Type 6, and (g) Type 7. The location of the Beijing metropolitan area is outlined in blue near the center of each panel. The black lines cutting through the Beijing metropolitan area in (a-g) indicate the location of the cross-sections shown in Fig. 11.





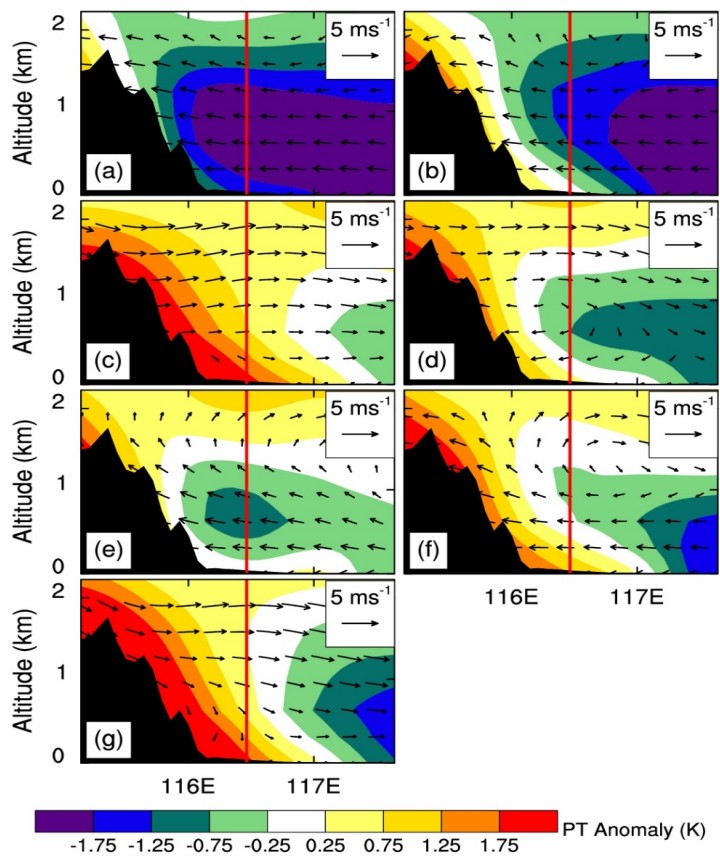

**Fig. 11.** Vertical cross-sections of the PT anomaly (subtracted from the average PT over the whole study region at each vertical layer, colored areas) and wind vectors (arrows) for the seven synoptic pattern types: (a) Type 1, (b) Type 2, (c) Type 3, (d) Type 4, (e) Type 5, (f) Type 6, and (g) Type 7. Note that before making the vertical cross-sections, the PT and wind vectors were interpolated to finer grids (0.1° × 0.1° grid spacing) using the *metgrid* module of the Weather Research and Forecasting model. The vertical wind component is multiplied by a factor of 30 when plotting the wind vector fields. The black area on the left side of each panel shows the terrain. The red line shows the location of the sounding station in Beijing. The location of the cross-section is shown by the black line in Fig. 10a.





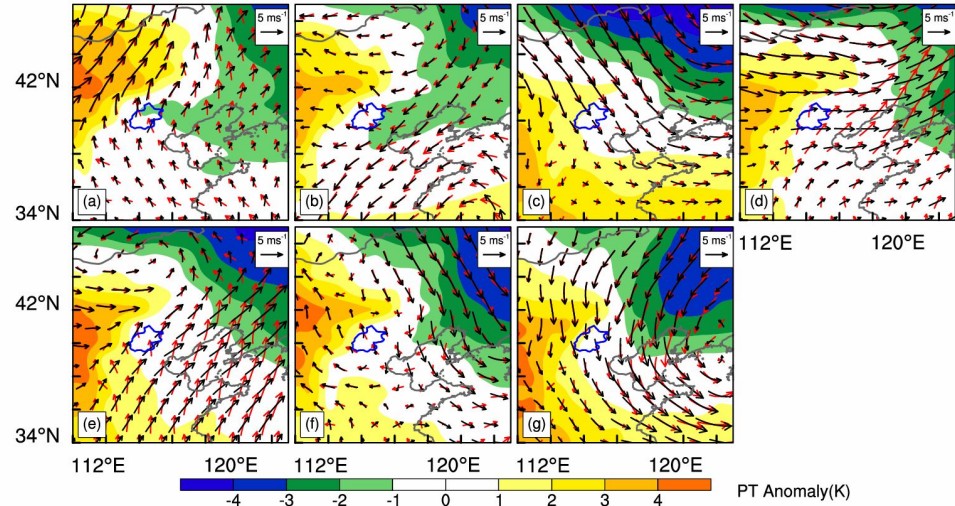

**Fig. 12.** Spatial distributions of the 800-hPa PT anomaly (subtracted from the average PT over the whole study region, colored areas) and wind vectors (black arrows) for the seven synoptic pattern types: (a) Type 1, (b) Type 2, (c) Type 3, (d) Type 4, (e) Type 5, (f) Type 6, and (g) Type 7. Red arrows show the 925-hPa wind field. The location of the Beijing metropolitan area is outlined in blue near the center of each panel.



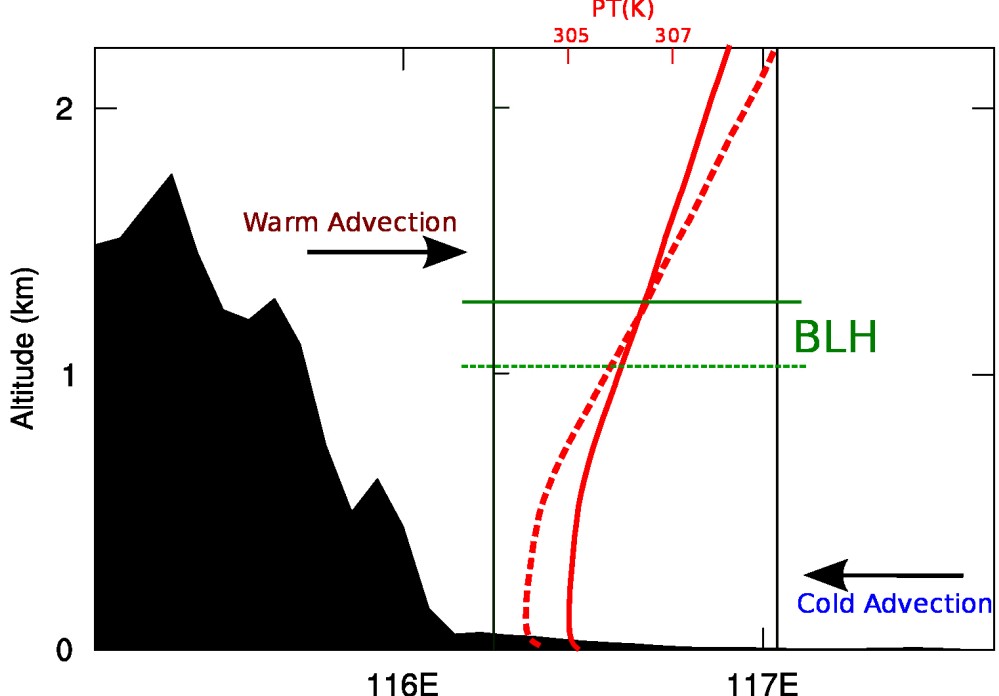

**Fig. 13.** Schematic diagram of the influence of near-surface cold advection and warm advection aloft on the BLH in Beijing in the afternoon during summer. The solid red line shows the vertical profile of PT in Beijing without any advection and the horizontal solid green line shows the corresponding BLH. The dashed red line shows the PT profile affected by both warm and cold advection and the horizontal dashed green line shows the top level of the PBL.