# Peer review of "Classification of summertime synoptic patterns in Beijing and their associations with boundary layer structure affecting aerosol pollution"

_Atmospheric Chemistry and Physics, 2016_

## Referee Comment (RC1) · Anonymous Referee #1 · 20 Dec 2016

This paper investigates the role of different weather conditions in modulating summertime boundary layer and further surface pollution in Beijing. A novel technique is applied to classify summertime circulation patterns into seven major weather types. More importantly, the authors thoroughly investigated the mechanism how different synoptic conditions impact on the variability of boundary layer height and structure, which is the major factor governing the vertical transport of pollutants and thus surface pollution level. The paper is also well presented and logically organized. I think this paper addresses an important issue and makes great contribution to our understanding of summer haze in Beijing. I thus recommend the paper be accepted for publication in ACP only with several minor comments.

[Figure]

Specific comments:

1. The introduction seems too long (4 pages). I think it could be shortened with an emphasize on the relationship between summer time boundary layer height and pollution in Beijing, and the novel method and data used in this study. 2. Section 2.3: While the T-mode PCA is conventionally used in classification, the oblique rotation may not be familiar to the major audience. It's better to elaborate here with some mathematical explanation of this technique. 3. Section 3.2: The authors identified seven dominant weather types. I wonder how is the order (No. 1-7) of these seven types determined? In traditional PCA, the modes are usually ordered by the amount of variance explained. Here it does not seem to be the case as their frequencies of occurrence are not ordered from high to low? Perhaps some explanation would help? 4. Also in this section, as types 1, 4 and 5 are associated with heavy pollution, I wonder if a composite analysis of these three types and the other four types would help better distinguish between their different characteristics in meteorological variables (RH, BLH, CLD, etc) and PM2.5 concentration? 5. Previously, there are also studies on the relationship between synoptic (circulation) patterns and pollution over the Beijing area, such as those cited by the authors in the introduction section. I wonder how is the current study compared with these previous results? Some discussions in a general context would help. 6. Figure 9 caption: "seasonally" should be "summer" because only summer data is analyzed here.

Typos:

Page 2, line 23: in the further suppression Page 3, line 25: the past decades Page 4, line 5: On local scale, or "On regional scale" Page 4, line 15: impacts on Page5, line 25: which subjectively defines a priori,. . . and in which the case assignment Page 9, line 2: ...PCA which is in the S-mode. . . Page 11, line 12: the lifting condensational level can drop. . . Page 12, line 3: understanding the effects of . . . Page 12, line 7: As illustrated in Fig. 8, . . . Page 12, line 18: well captured Page 12, line 2d: over the ENTIRE study region Page 12, line 26: impose a negative thermal anomaly ON the

PBL there Page 13, line 5: leading to THE suppression of . . .

---

## Referee Comment (RC2) · Anonymous Referee #2 · 22 Dec 2016

This study utilizes four-year sounding measurements, surface PM measurements and reanalysis data to examine the influence of the synoptic patterns on the planetary boundary layer (PBL) structure and air pollution in Beijing. As Beijing has been experiencing extremely severe particulate pollution in the past few years, this study shed light on the contribution of the regional scale dynamics to the haze formation in a quantitative way. Using a synoptic pattern classification method, three patterns are identified to be closely related with heavy pollution condition in Beijing and the underlying dynamical processes are revealed in details. The cloud influence on PBL is also assessed. Overall, the manuscript is well written and I recommend publishing this study on ACP after some minor questions below can be addressed by the authors.

1. Page 3, Line 9-19. I am kind of surprised that the contribution from the automobile exhaust to the Beijing air pollution was not even mentioned. Some associated references can be added here (Zhang et al., 2015, Chem. Rev.; Peng et al., 2016, PNAS)

2. Page 4, Line 19. Please be specific on how the sea-breeze affects PBL and if it alleviates or deteriorates air pollution.

3. Page 10, Line 11. I don't understand how come R is low as -0.37 but p-value is less than 0.01. What significance test is performed here?

4. Section 3.2. Does a haze event have to be tied to a synoptic pattern? How about a 'no wind' condition? It seems not belonging to none of the seven synoptic patterns listed there, but it did occur during some severe haze event.

5. Fig. 13. The schematic diagram is very interesting, but the mechanism only works for daytime. The pollution is typically even worse during the nighttime. It will be interesting to have some discussion about the possible PBL-synotptic pattern interactions during the nighttime.

---

## Referee Comment (RC3) · Anonymous Referee #3 · 26 Dec 2016

**Comments on "Classification of summertime synoptic patterns in Beijing and their association with boundary layer structure affecting aerosol pollution"**

**General Comments**

It is well known that air pollution is directly associated with atmospheric boundary layer height (BLH). The daytime convective boundary layer (CBL) develops in the synoptic background. Thus the synoptic conditions affect the BLH and consequently air pollution. In this paper, the authors divided the summertime synoptic conditions in Beijing area into seven typical patterns and analyzed the BLH and air pollution level in different synoptic patterns. The results suggest that, the positive synoptic conditions promote CBL development, and higher BLH leads to light air pollution, whereas the adverse synoptic conditions suppress CBL development, and lower BLH leads to heavy air pollution. The authors provided some details about how the special synoptic conditions influence the BLH, and proposed a possible mechanism to explain the reason. The results in this paper can help to understand the impact of synoptic conditions on air pollution. However, some statements and discussion are not convincing, and the English writing should be further improved. Therefore, my recommendation is publication in ACP after major revisions.

**Specific Comments**

1. For Eq. (1), presence of u$_*$ (not $bu_*^2$) in the right hand side is an error. If the authors used this formula to calculate the BLH, the results are incorrect.

2. This study emphasizes that heavy air pollution is caused by low BLH. But the authors did not provide solid evidences. In Fig. 3b, the results show that the diurnal variation of 1 h-bin averaged PM2.5 concentration is not significant, but the difference in BLH between 14:00 and 08:00 (or 20:00) is very larger. In addition, even for the situation at 14:00, Fig. 4b shows that the correlation coefficient between PM2.5 concentration and BLH is relatively low. Then the problem arises. What is the major reason for the formation of heavy air pollution in Beijing in summertime, reduction of BLH or transportation of pollutant? In my opinion, discussing the impact of BLH on air pollution level is based on the premise that the air pollution is caused by local emissions. The above mentioned results suggest it may be not the case. If the air pollution is caused by transportation of pollutant, the low BLH may be the result rather than the reason of heavy air pollution. Therefore, the authors should be cautious when discussing the relationship between BLH and air pollution level, and state their results more reasonably.

3. For the results in Fig. 8, I do not know how the authors obtained the correlation coefficients. This figure shows that the correlation coefficient between PM2.5

concentration and BLH is -0.97 (the absolute value is very close to 1.0). In page 12 line 7, the authors said 'the BLH is the most crucial factor related to aerosol pollution level under different synoptic conditions'. But Fig. 4b shows that the correlation coefficient is very low (the absolute value is smaller than 0.4). I cannot understand such a large difference between the two results. The authors should explain why. Secondly, Fig. 8 shows a high positive correlation between PM2.5 concentration and CLD and a high negative correlation between PM2.5 concentration and CLD. These results imply that the lower BLH is highly related to the larger cloud cover. It is know that the daytime CBL is driven by surface heating. The large cloud cover reduces radiation arriving the surface and then the surface sensible heat flux. This may be the reason for the lower BLH in cloudy days. But the authors only emphasize the effect of capping inversion (they propose a mechanism about this as illustrated in Fig. 13). Thirdly, results in this figure suggest that the wind direction plays an important role in the formation of air pollution. High air pollution level is associated with south wind, implying that the pollutants may come from the cities south of Beijing. In this situation the lower BLH may be caused by the enhanced air pollution. However, the authors' analyses give me a strong feeling that the reduced BLH leads to heavy air pollution. So my question is how to interpret the results in Fig. 8. The authors should provide us a "clear picture".

4. Page 13, line 1-2 'Among the seven identified synoptic patterns, the strongest near-surface could advection is associated with Type 1 (Fig. 11a), leading to the coldest PBL at 1400 BJT (Fig. 9a)', and line 5-6 'Types 2, 4, 5 and 6 also show cold advection toward Beijing but it is less prominent (Figs. 11b and 11d-f)'. I am not sure if the PT anomaly is caused by cold advection. Large cloud cover may also reduce PT in the boundary layer. The PT anomaly in Type 2 is similar to that in Type 1, and the other conditions in the two types are almost the same: the same CLD, no warm advection above CBL top. Why Type 1 has a negative BLH anomaly but Type 2 has a positive BLH anomaly? Moreover, the BLH in Type 1 is slightly lower than the seasonal average while the BLH in Type 2 is slightly higher than the seasonal average (as shown in Fig. 6a), and the BLH difference in the two types is merely about 200 m. Why such a small difference in BLH can introduce large difference in PM2.5 concentration (one is 101 $\mu g\ m^{-3}$, another is 67$\mu g\ m^{-3}$)? I guess, transportation of high concentration pollutants may contribute to heavy air pollution in Type1, because Type 1 has south wind whereas Type 2 has east wind.

5. For Fig. 11, I do not think the PT anomaly and the wind field can match very well. Fig. 11d shows an elevated negative PT-anomaly area, stretching from the right to the left. But the wind direction is form the left to the right together with a downward component. Actually, Fig. 6a shows that the boundary layer wind blows towards northeast (with a relatively small west component). It means that the flow passing Beijing does not come from Bohai or Yellow Sea (the same

evidence can be found in Fig. 10d). Also, Fig. 11e shows an isolated maximum negative PT-anomaly area over Beijing. If this negative PT-anomaly area is caused by cold advection, the magnitude of PT-anomaly in the right area should be larger than, or at least the same as, that in this area. I mean the maximum negative PT-anomaly area should stretch to the right side of picture, as shown in Figs. 11a&b or Fig. 11f. Can the isolated maximum negative PT-anomaly area over land be regarded as the result of cold advection from sea? In my opinion, the isolated maximum negative PT-anomaly area over Beijing implies a local cooling. So, my question is, can the negative PT-anomaly be interpreted as the result of cold advection from sea? I think the authors should discuss this issue cautiously.

6.  For Fig. 13, I suggest to remove this schematic map. I think there is no solid evidence to support the so-called "advection mechanism". The authors can add the seasonal mean PT profile in each panel of Fig. 9. By comparing the PT profile in each type with the seasonal mean PT profile, we can know whether the capping inversion is enhanced or weakened in each synoptic pattern. Then the authors can discuss the possible reasons.

7.  For the English wording and writing, I suggest that the authors get a fluent writer/speaker of English to look through the paper.

---

## Author Comment (AC1) · 23 Jan 2017

**Authors' Response to Referees' Comments**

**Anonymous Reviewer #1**

This paper investigates the role of different weather conditions in modulating summertime boundary layer and further surface pollution in Beijing. A novel technique is applied to classify summertime circulation patterns into seven major weather types. More importantly, the authors thoroughly investigated the mechanism how different synoptic conditions impact on the variability of boundary layer height and structure, which is the major factor governing the vertical transport of pollutants and thus surface pollution level. The paper is also well presented and logically organized. I think this paper addresses an important issue and makes great contribution to our understanding of summer haze in Beijing. I thus recommend the paper be accepted for publication in ACP only with several minor comments.

First of all, we appreciate tremendously the reviewer's positive comments. In response to the reviewer's comments, we have made relevant revisions to the manuscript. Listed below are our responses and the corresponding changes made to the manuscript according to suggestions given by the reviewer. Each comment of the reviewer (in black) is listed, followed by our responses (in blue).

Specific comments:

1. The introduction seems too long (4 pages). I think it could be shortened with an emphasize on the relationship between summer time boundary layer height and pollution in Beijing, and the novel method and data used in this study.

Per your suggestion, the description was deleted with regard to the relationship between summer time boundary layer height and pollution in Beijing. Besides, several important literatures were added to the introduction as suggested by reviewer #2 to give a more comprehensive overview of current research and highlight the importance of this study as well.

2. Section 2.3: While the T-mode PCA is conventionally used in classification, the oblique rotation may not be familiar to the major audience. It's better to elaborate here with some mathematical explanation of this technique.

Per your suggestion, the following mathematical explanation with regard to T-mode PCA were added in the revised manuscript:

"To speed up the calculation of PCA, the data is split into ten subsets; and then, the principle components (PCs) obtained from each subset are projected on the rest data. The T-PCA classification based on cost733 software package includes the following steps:

(1) The data is standardized spatially. Each pattern's mean is subtracted from the data, and then the patterns are divided by their standard deviations.

(2) The data is split into ten subsets through selecting the data once every ten days. For example, the first subset consists of the 1st, 11th, 21st, 31st, etc. days, and the second subset consists of the 2nd, 12th, 22nd, 32nd, etc. days.

(3) The PCs are calculated using the singular value decomposition for each subset. And the PCs of each subset are ordered according to the magnitude of their explained variances.

(4) An oblique rotation (using direct oblimin) is applied on the PCs, employing an adaptation of the Gradient Projection Algorithm of Bernaards and Jennrich (2005). The main reason for using rotation is to facilitate the interpretation (Abdi and Williams, 2010). This transformation does not constrain the orthogonality, allowing for the PCs the freedom to better reflect the original data (Richman, 1981).

(5) The PC scores of each subset are projected onto the remaining data by solving the matrix equation: $\Phi A^T = F^T Z$, where F and $\Phi$ are matrices of PC scores and PC correlations, respectively, and Z is the full data matrix, and A are pseudo-loadings to be determined. Each day is classified with the PC (type) for which it has the highest loading.

(6) Contingency tables are finally used to compare the ten classifications, and the classification most consistent with the other nine classifications is selected as the

resultant one."

3. Section 3.2: The authors identified seven dominant weather types. I wonder how is the order (No. 1-7) of these seven types determined. In traditional PCA, the modes are usually ordered by the amount of variance explained. Here it does not seem to be the case as their frequencies of occurrence are not ordered from high to low? Perhaps some explanation would help?

Actually, the seven synoptic types (#1-7) were also determined by the amount of variance explained, which were 0.192415, 0.112347, 0.106, 0.092, 0.085, 0.079, and 0.077, respectively. To better understand the T-PCA, the detailed classification procedures were added on Page 9-10 in our revised manuscript (see our response to comment #2).

4. Also in this section, as types 1, 4 and 5 are associated with heavy pollution, I wonder if a composite analysis of these three types and the other four types would help better distinguish between their different characteristics in meteorological variables (RH, BLH, CLD, etc) and PM$_{2.5}$ concentration?

In the revised manuscript, we compared the meteorological variables and aerosol concentration of polluted synoptic types (1, 4, and 5) with that of other types (2, 3, 6, and 7), which were detailed in the newly added Table S2 in Supplementary Materials. It is noteworthy that the heavy aerosol pollution of Types 1, 4, and 5 are associated with the relatively low BLH, high RH2, high CLD, and high frequency of southerly PBL winds.

The relevant discussion was added on Page 13 in the revised manuscript.

*Table S2. Statistics of the correlation between PM2.5 concentrations and meteorological variables corresponding to the polluted synoptic types (1, 4, and 5) and other synoptic types. The meteorological variables (mean values ± one standard deviation) include 2-m temperature (T2), 2-m relative humidity (RH2), wind speed at the 925-hPa level (WS), southerly wind frequencies at the 925h-hPa level (WD), total cloud cover at 1400 BJT (CLD), and the BLH at 1400 BJT. The correlation coefficients (R) between the meteorological variable and PM2.5 concentration are also given, which are calculated based on the seven pairs of mean values for each*

*synoptic pattern.*

| # | Polluted types (1, 4, and 5) | Other types (2, 3, 6, and 7) | R (#, PM$_{2.5}$) |
|---|---|---|---|
| PM$_{2.5}$ (µg m$^{-3}$) | 99.7 ± 51.9 | 56.8 ± 40.2 | / |
| BLH (km) | 1.17 ± 0.59 | 1.63 ± 0.69 | -0.97* |
| RH2 (%) | 68.1 ± 14.3 | 59.4 ± 14.1 | 0.85* |
| CLD (%) | 92 ± 22 | 74 ± 36 | 0.84* |
| 925-hPa WD (South) (%) | 61 ± 6 | 49 ± 7 | 0.78* |
| T2 (K) | 299.5 ± 2.5 | 299.8 ± 2.7 | -0.33 |
| 925-hPa WS (m s$^{-1}$) | 5.1 ± 3.1 | 4.9 ± 3.3 | -0.02 |

5. Previously, there are also studies on the relationship between synoptic (circulation) patterns and pollution over the Beijing area, such as those cited by the authors in the introduction section. I wonder how the current study compared with these previous results. Some discussions in a general context would help.

The following discussions were added on Page 13:

"The relationship between synoptic patterns/circulations and pollution in Beijing unraveled in this study is similar to that of previous studies of Zhang et al. (2012) and Ye et al. (2016). When high pressure located to the east/southeast of Beijing, the resultant southerly PBL winds would bring the pollutants emitted from the southern Hebei to Beijing."

6. Figure 9 caption: "seasonally" should be "summer" because only summer data is analyzed here.

Amended as suggested.

Typos:

Page 2, line 23: in the further suppression

Page 3, line 25: the past decades

Page 4, line 5: On local scale, or "On regional scale"

Page 4, line 15: impacts on

Page5, line 25: which subjectively defines a priori … and in which the case assignment

Page 9, line 2: ...PCA which is in the S-mode …

Page 11, line 12: the lifting condensational level can drop …

Page 12, line 3: understanding the effects of …

Page 12, line 7: As illustrated in Fig. 8, …

Page 12, line 18: well captured

Page 12, line 2d: over the ENTIRE study region

Page 12, line 26: impose a negative thermal anomaly ON the PBL there

Page 13, line 5: leading to THE suppression of …

All the typos have been revised as suggested.

**References:**

Abdi, H. and Williams, L. J.: Principal component analysis, Wiley Interdiscip. Rev. Comput. Stat., 2(4), 433–459, doi:10.1002/wics.101, 2010.

Bernaards, C. A. and Jennrich, R. I.: Gradient projection algorithms and software for arbitrary rotation criteria in factor analysis, Educ. Psychol. Meas., 65(5), 676–696, doi:10.1177/0013164404272507, 2005.

Richman, M. B.: Obliquely rotated principal components: An improved meteorological map typing technique?, J. Appl. Meteorol., 20(10), 1145–1159, doi:10.1175/1520-0450(1981)020<1145:ORPCAI>2.0.CO;2, 1981.

Ye, X., Song, Y., Cai, X., and Zhang, H.: Study on the synoptic flow patterns and boundary layer process of the severe haze events over the North China Plain in January 2013, Atmos. Environ., 124(January 2013), 129–145, doi:10.1016/j.atmosenv.2015.06.011, 2016.

Zhang, J. P., Zhu, T., Zhang, Q. H., Li, C. C., Shu, H. L., Ying, Y., Dai, Z. P., Wang, X., Liu, X. Y., Liang, A. M., Shen, H. X., and Yi, B. Q.: The impact of circulation patterns on regional transport pathways and air quality over Beijing and its surroundings, Atmos. Chem. Phys., 12(11), 5031–5053, doi:10.5194/acp-12-5031-2012, 2012.

---

## Author Comment (AC2) · 23 Jan 2017

**Authors' Response to Referees' Comments**

**Anonymous Reviewer #2**

This study utilizes four-year sounding measurements, surface PM measurements and reanalysis data to examine the influence of the synoptic patterns on the planetary boundary layer (PBL) structure and air pollution in Beijing. As Beijing has been experiencing extremely severe particulate pollution in the past few years, this study shed light on the contribution of the regional scale dynamics to the haze formation in a quantitative way. Using a synoptic pattern classification method, three patterns are identified to be closely related with heavy pollution condition in Beijing and the underlying dynamical processes are revealed in details. The cloud influence on PBL is also assessed. Overall, the manuscript is well written and I recommend publishing this study on ACP after some minor questions below can be addressed by the authors.

We thank reviewer #2 for his/her positive comments on our manuscript. In response to his/her comments, we have made relevant revisions to the manuscript. Listed below are our responses and the corresponding changes made to the manuscript according to suggestions given by the reviewer. Each comment (in black) is listed, followed by our responses (in blue).

1. Page 3, Line 9-19. I am kind of surprised that the contribution from the automobile exhaust to the Beijing air pollution was not even mentioned. Some associated references can be added here (Zhang et al., 2015, Chem. Rev.; Peng et al., 2016, PNAS)

Per your kind suggestions, the literatures concerning the contribution from automobile exhaust were added as below:

"*The major sources of aerosol in Beijing include traffic emission, power plant, industry, domestic emission, and agricultural activities (R. Zhang et al., 2015; Liu et al., 2016; Peng et al., 2016).*"

2. Page 4, Line 19. Please be specific on how the sea-breeze affects PBL and if it alleviates or deteriorates air pollution.

The impacts of sea-breeze on PBL structure and air pollution were added on Page 4 in the revised manuscript, as shown below.

*"The diurnal variation of land-breeze and sea-breeze provides a mechanism for the pollutants in the Beijing-Tianjin-Hebei region to be recirculated and accumulated. In the evening and early morning, the presence of land-breeze (offshore wind) could bring the pollutants emitted from coastal regions to Bohai sea, and then in the afternoon, the development of sea-breeze (onshore wind) could bring these pollutants back to coastal regions, leading to exacerbated pollution. With the sea-breeze penetrates further inland, the pollutants emitted from coastal regions could be transported to the downstream regions."*

3. Page 10, Line 11. I don't understand how come R is low as -0.37 but p-value is less than 0.01. What significance test is performed here?

The Pearson correlation coefficient (R) measures the linear relationship between two dataset. The closer to 1 the more 'confident' we are of a positive linear correlation and the closer to -1 the more 'confident' we are of a negative linear correlation.

The confidence in a correlation is formally determined not just by the correlation coefficient but also by the number of pairs in your data. If there are very few pairs then R needs to be very close to 1 or -1 for it to be deemed 'statistically significant', but if there are many pairs then a R closer to 0 can still considered 'highly significant' (Fenton and Neil., 2012). The critical values of R could be found at following website (Table B.7): onlinelibrary.wiley.com/doi/10.1002/9781118342978.app2/pdf

One of the standard methods to measure the 'significance' is the p-value, which is a number between 0 and 1 representing the probability that the data would have arisen if the null hypothesis (i.e., the slope of the regression line is zero) were true. In this study, there are 282 pairs of data whose correlation coefficient is -0.37, and the p-value is less than 0.01. This means the chance that we would have seen these data

pairs were unrelated is less than 1%.

In this study, the p-value was calculated based on the student's t-distribution (Weathington et al., 2012), which was added in the caption of Figure 4.

4. Section 3.2. Does a haze event have to be tied to a synoptic pattern? How about a 'no wind' condition? It seems not belonging to none of the seven synoptic patterns listed there, but it did occur during some severe haze event.

Yes, every haze event develops under a specific synoptic condition, even for those haze episodes under "no wind" condition. Table S1 (in the Supplementary Materials) shows the synoptic type for each day of time period (2011 to 2014) investigated. A large body of literatures (e.g., Chen et al., 2008; Wei et al., 2011; Zhang et al., 2012) has demonstrated that a synoptic pattern is an important factor modulating the day-to-day variation of air quality. Also, the near-surface calm wind situations are often linked to a certain synoptic forcing (Kim Oanh and Leelasakultum, 2011; Ye et al., 2016), although the impacts of synoptic pattern on calm wind situations may be relatively weak, which still cannot be ignored.

5. Fig. 13. The schematic diagram is very interesting, but the mechanism only works for daytime. The pollution is typically even worse during the nighttime. It will be interesting to have some discussion about the possible PBL-synoptic pattern interactions during the nighttime.

We totally agree with the reviewer, the nocturnal boundary layer (NBL) structures/processes also play an important role in modulating the air pollution. During a diurnal cycle, the relatively high nighttime concentration of pollutants is primarily induced by the drop of BLH after sunset and the variation of emission. Besides, the daytime PBL processes also partition the pollutants within NBL and those retained in the residual layer aloft. Therefore, the following discussion has been added in the Conclusions.

"*Although this study focuses on the daytime PBL structure, the nocturnal PBL also significantly affects the air quality at hourly to diurnal scales through the*

*intermittent turbulence, which also cannot be ignored. The structure of nocturnal PBL is primarily determined by (stull, 1988; Salmond and Mckendry, 2005). The nocturnal PBL may range from fully turbulent to intermittently turbulent or even non-turbulent at a variety of heights, temporal scales and spatial locations, which was largely induced by complex interactions between the static stability of the atmosphere and those processes (i.e. wind shear from synoptic patterns, terrain induced flows, low-level jets) that govern mechanical generation of turbulence. This makes it very difficult for the observation of large-scale atmospheric advection, investigation of the PBL-synoptic pattern interaction, transport pathways and dispersion of pollutants in the NBL, particular in regions of complex terrain such as Beijing. To fully understand the impacts of PBL on air quality in Beijing, more attention should be paid to the nocturnal PBL in the future."*

**References:**

Chen, Z. H., Cheng, S. Y., Li, J. B., Guo, X. R., Wang, W. H. and Chen, D. S.: Relationship between atmospheric pollution processes and synoptic pressure patterns in northern China, Atmos. Environ., 42(24), 6078–6087, doi:10.1016/j.atmosenv.2008.03.043, 2008.

Fenton, N. and Neil M.: Risk Assessment and Decision Analysis with Bayesian Networks. CRC Press, 2012.

Liu, Z., Wang, Y., Hu, B., Ji, D., Zhang, J., Wu, F., Wan, X. and Wang, Y.: Source appointment of fine particle number and volume concentration during severe haze pollution in Beijing in January 2013, Environ. Sci. Pollut. Res., 23(7), 6845–6860, doi:10.1007/s11356-015-5868-6, 2016.

Kim Oanh, N. T. and Leelasakultum, K.: Analysis of meteorology and emission in haze episode prevalence over mountain-bounded region for early warning, Sci. Total Environ., 409(11), 2261–2271, doi:10.1016/j.scitotenv.2011.02.022, 2011.

Peng, J., Hu, M., Guo, S., Du, Z., Zheng, J., Shang, D., Levy Zamora, M., Zeng, L., Shao, M., Wu, Y.-S., Zheng, J., Wang, Y., Glen, C. R., Collins, D. R., Molina, M. J. and Zhang, R.: Markedly enhanced absorption and direct radiative forcing of black carbon under polluted urban environments, Proc. Natl. Acad. Sci., 113(16), 4266–4271, doi:10.1073/pnas.1602310113, 2016.

Weathington, B. L., Cunningham, C. J. L, and Pittenger, D. J.: Understanding

Business Research. John Willey & Sons, 2012.

Wei, P., Cheng, S., Li, J. and Su, F.: Impact of boundary-layer anticyclonic weather system on regional air quality, Atmos. Environ., 45(14), 2453–2463, doi:10.1016/j.atmosenv.2011.01.045, 2011.

Ye, X., Song, Y., Cai, X., and Zhang, H.: Study on the synoptic flow patterns and boundary layer process of the severe haze events over the North China Plain in January 2013, Atmos. Environ., 124, 129–145, doi:10.1016/j.atmosenv.2015.06.011, 2016.

Zhang, J. P., Zhu, T., Zhang, Q. H., Li, C. C., Shu, H. L., Ying, Y., Dai, Z. P., Wang, X., Liu, X. Y., Liang, A. M., Shen, H. X., and Yi, B. Q.: The impact of circulation patterns on regional transport pathways and air quality over Beijing and its surroundings, Atmos. Chem. Phys., 12(11), 5031–5053, doi:10.5194/acp-12-5031-2012, 2012.

Zhang, R., Wang, G., Guo, S., Zamora, M. L., Ying, Q., Lin, Y., Wang, W., Hu, M. and Wang, Y.: Formation of Urban Fine Particulate Matter, Chem. Rev., 115(10), 3803–3855, doi:10.1021/acs.chemrev.5b00067, 2015.

---

## Author Comment (AC3) · 23 Jan 2017

**Authors' Response to Referees' Comments**

**Anonymous Reviewer #3**

It is well known that air pollution is directly associated with atmospheric boundary layer height (BLH). The daytime convective boundary layer (CBL) develops in the synoptic background. Thus the synoptic conditions affect the BLH and consequently air pollution. In this paper, the authors divided the summertime synoptic conditions in Beijing area into seven typical patterns and analyzed the BLH and air pollution level in different synoptic patterns. The results suggest that, the positive synoptic conditions promote CBL development, and higher BLH leads to light air pollution, whereas the adverse synoptic conditions suppress CBL development, and lower BLH leads to heavy air pollution. The authors provided some details about how the special synoptic conditions influence the BLH, and proposed a possible mechanism to explain the reason. The results in this paper can help to understand the impact of synoptic conditions on air pollution. However, some statements and discussion are not convincing, and the English writing should be further improved. Therefore, my recommendation is publication in ACP after major revisions.

First of all, we appreciate the reviewer's comments and suggestions. In response to the reviewer's comments, we have made thorough revisions to the manuscript. Listed below are our responses and the corresponding changes made to the manuscript according to suggestions given by the reviewer. Each comment (in black) is listed, followed by our responses (in blue).

**Specific Comments**

1. For Eq. (1), presence of u* in the right hand side is an error. If the authors used this formula to calculate the BLH, the results are incorrect.

This was a typo for Eq. (1), which has been corrected in the revised manuscript.

2. This study emphasizes that heavy air pollution is caused by low BLH. But the

authors did not provide solid evidences. In Fig. 3b, the results show that the diurnal variation of 1 h-bin averaged PM2.5 concentration is not significant, but the difference in BLH between 14:00 and 08:00 (or 20:00) is very larger. In addition, even for the situation at 14:00, Fig. 4b shows that the correlation coefficient between PM2.5 concentration and BLH is relatively low. Then the problem arises. What is the major reason for the formation of heavy air pollution in Beijing in summertime, reduction of BLH or transportation of pollutant? In my opinion, discussing the impact of BLH on air pollution level is based on the premise that the air pollution is caused by local emissions. The above mentioned results suggest it may be not the case. If the air pollution is caused by transportation of pollutant, the low BLH may be the result rather than the reason of heavy air pollution. Therefore, the authors should be cautious when discussing the relationship between BLH and air pollution level, and state their results more reasonably.

We totally agree with the reviewer, the BLH is just one of the meteorological factors modulating aerosol pollution. In the revised manuscript, the relationships between BLH and aerosol pollution were rewritten to make the explanation more reasonably. In addition, either horizontal transport of pollutants or aerosol-PBL feedback plays important roles in modulating the air quality under different synoptic conditions. Therefore, the relevant discussions were added on Page 13 and Page 17 in the revised manuscript.

3.1 For the results in Fig. 8, I do not know how the authors obtained the correlation coefficients. This figure shows that the correlation coefficient between PM2.5 concentration and BLH is -0.97 (the absolute value is very close to 1.0). In page 12 line 7, the authors said 'the BLH is the most crucial factor related to aerosol pollution level under different synoptic conditions'. But Fig. 4b shows that the correlation coefficient is very low (the absolute value is smaller than 0.4). I cannot understand such a large difference between the two results. The authors should explain why.

The correlation coefficients in Fig.8 were calculated based on the mean values for each synoptic type. For example, the correlation of BLH and $PM_{2.5}$ concentration was

calculated using the seven pairs of BLH and PM$_{2.5}$ concentration. In contrast, the correlation coefficients in Fig.4 were directly calculated using daily averaged PM$_{2.5}$ concentration and its concomitant BLH. In the revised manuscript, such information was added on the caption of Fig. 8.

3.2 Secondly, Fig. 8 shows a high positive correlation between PM2.5 concentration and BLH and a high negative correlation between PM2.5 concentration and CLD. These results imply that the lower BLH is highly related to the larger cloud cover. It is know that the daytime CBL is driven by surface heating. The large cloud cover reduces radiation arriving the surface and then the surface sensible heat flux. This may be the reason for the lower BLH in cloudy days. But the authors only emphasize the effect of capping inversion (they propose a mechanism about this as illustrated in Fig. 13).

We totally agree with the reviewer that the effect of cloudiness cannot be ignored in modulating the BLH, which should not be de-emphasized. In addition to the cloudiness, the cool/warm advections associated with different synoptic patterns may also affect the PBL structure. In the revised manuscript, to better understand the effects of cool/warm advections on PBL structure, several idealized numerical experiments were conducted.

In the idealized experiments, the simulation region was set as the same studied region shown in Fig. 1, with a horizontal grid spacing of 0.1° (~11 km). In the vertical dimension, 48 layers were set from the surface to 100-hPa level, with 21 layers between the surface and 2 km above ground level (AGL). To isolate the effects of synoptic forcing (e.g. warm/cold advection) from cloudiness, the microphysics and cumulus schemes were turned off in the idealized simulations. Similar approaches have been used by De Wekker (2008) to investigate the suppression of BLH near a mountain, and by Pu and Dickinson (2014) to investigate the dynamics of Low-Level Jet over the Great Plains.

In the revised manuscript, the description of numerical experiments were added on Page 10 to 11. And the relevant results and discussions were added on Page 15 to

3.3 Thirdly, results in this figure suggest that the wind direction plays an important role in the formation of air pollution. High air pollution level is associated with south wind, implying that the pollutants may come from the cities south of Beijing. In this situation the lower BLH may be caused by the enhanced air pollution. However, the authors' analyses give me a strong feeling that the reduced BLH leads to heavy air pollution. So my question is how to interpret the results in Fig. 8. The authors should provide us a "clear picture".

As the anti-correlated relationship illustrate in Fig. 4 and Fig. 8, the reduced BLH may be one factor modulating the pollution level, however, the horizontal transport of pollutants should not be de-emphasized as the reviewer suggested. In the revised manuscript, we try to provide a more "clear picture" about the relationships between meteorological factors/processes (i.e., BLH, horizontal transportation, aerosol-PBL feedback) and pollution. The relevant discussions were added on Page 13 to 14.

4. Page 13, line 1-2 'Among the seven identified synoptic patterns, the strongest near-surface could advection is associated with Type 1 (Fig. 11a), leading to the coldest PBL at 1400 BJT (Fig. 9a)', and line 5-6 'Types 2, 4, 5 and 6 also show cold advection toward Beijing but it is less prominent (Figs. 11b and 11d-f)'. I am not sure if the PT anomaly is caused by cold advection. Large cloud cover may also reduce PT in the boundary layer. The PT anomaly in Type 2 is similar to that in Type 1, and the other conditions in the two types are almost the same: the same CLD, no warm advection above CBL top. Why Type 1 has a negative BLH anomaly but Type 2 has a positive BLH anomaly? Moreover, the BLH in Type 1 is slightly lower than the seasonal average while the BLH in Type 2 is slightly higher than the seasonal average (as shown in Fig. 6a), and the BLH difference in the two types is merely about 200 m. Why such a small difference in BLH can introduce large difference in $PM_{2.5}$ concentration (one is 101 µg m-3, another is 67µg m-3)? I guess, transportation of high concentration pollutants may contribute to heavy air pollution in Type1, because

Type 1 has south wind whereas Type 2 has east wind.

We agree with the reviewer, the large difference of PM$_{2.5}$ concentration between Type 1 and Type 2 may be primarily caused by the different horizontal transport of pollutants. The relevant discussions were added on Page 13 in the revised manuscript.

Besides, to understand the impacts of cool/warm advection on PBL structure, several idealized simulations were conducted to isolate the impacts of advection from other factors, such as the cloudiness and aerosol-PBL feedback. As the simulated cross section of PT shown in Fig. 11, the cold/warm advection could play a role in modulating the PBL structure in Beijing. The relevant discussions were added on Page 15-16 in the revised manuscript.

5. For Fig. 11, I do not think the PT anomaly and the wind field can match very well. Fig. 11d shows an elevated negative PT-anomaly area, stretching from the right to the left. But the wind direction is form the left to the right together with a downward component. Actually, Fig. 6a shows that the boundary layer wind blows towards northeast (with a relatively small west component). It means that the flow passing Beijing does not come from Bohai or Yellow Sea (the same evidence can be found in Fig. 10d). Also, Fig. 11e shows an isolated maximum negative PT-anomaly area over Beijing. If this negative PT-anomaly area is caused by cold advection, the magnitude of PT-anomaly in the right area should be larger than, or at least the same as, that in this area. I mean the maximum negative PT-anomaly area should stretch to the right side of picture, as shown in Figs. 11a&b or Fig. 11f. Can the isolated maximum negative PT-anomaly area over land be regarded as the result of cold advection from sea? In my opinion, the isolated maximum negative PT-anomaly area over Beijing implies a local cooling. So, my question is, can the negative PT-anomaly be interpreted as the result of cold advection from sea? I think the authors should discuss this issue cautiously.

We agree with the reviewer, the development of PBL can be extremely complex, influencing by the cloudiness, warm/cold advection, and aerosols at the same time. And the resolution of FNL reanalysis may be not enough to study the PBL

structure/process. Thus, in the revised manuscript, seven idealized numerical experiments were conducted to understand the effects of cold/warm advection on PBL structure, in which the effects of cloudiness were isolated through turning off relevant parameterization schemes. The relevant discussions were added on Page 15-16 in the revised manuscript.

6. For Fig. 13, I suggest to remove this schematic map. I think there is no solid evidence to support the so-called "advection mechanism". The authors can add the seasonal mean PT profile in each panel of Fig. 9. By comparing the PT profile in each type with the seasonal mean PT profile, we can know whether the capping inversion is enhanced or weakened in each synoptic pattern. Then the authors can discuss the possible reasons.

The schematic diagram was removed as suggested, and the revised Fig.9 was drawn, in which the PT anomaly (subtracted from the seasonally averaged PT profile in summer) for each synoptic type was added.

For Types 1, 4, and 5, the differential cooling/ warming anomalies within PBL and above it play a role in enhancing the thermal inversion at PBL top, which would suppress the development of PBL. Specifically, the cooling (warming) of Type 1 (4) is stronger (weaker) within PBL than that above it; and for Type 5, the PT anomaly within PBL is negative while that above PBL is positive. In contrast, the PT anomaly of other Types (2, 3, 6, and 7) tend to lower the thermal inversion at PBL top to some extent, favoring the growth of PBL. These PT anomalies and resultant BLHs could be partially responsible to the different pollution level for different synoptic types. In the revised manuscript, the relevant discussions were added on Page 15.

[Figure]

*Fig. 9. Vertical profiles of potential temperature (PT) at 1400 BJT associated with the seven types of synoptic pattern derived from soundings (red lines). Solid lines indicate average values and shaded areas show the uncertainty range (the mean ± one standard deviation). Green solid lines represent the summer averaged BLH and green dashed lines represent the average BLH for each synoptic type. The PT anomaly (subtracted from the summer averaged PT profile) for each synoptic type was also given by the blue dash-lines in each panel.*

7. For the English wording and writing, I suggest that the authors get a fluent writer/speaker of English to look through the paper.

Per your kind suggestion, a native speaker has been invited to review the manuscript.

**References:**

De Wekker, S. F. J.: Observational and numerical evidence of depressed convective boundary layer heights near a mountain base, J. Appl. Meteorol. Climatol., 47(4), 1017–1026, doi:10.1175/2007JAMC1651.1, 2008.

Pu, B. and Dickinson, R. E.: Diurnal spatial variability of Great Plains summer precipitation related to the dynamics of the low-level jet, J. Atmos. Sci., 71, 1807–1817, doi:10.1175/JAS-D-13-0243.1, 2014.